# Prenatal and Neonatal Bone Health: Updated Review on Early Identification of Newborns at High Risk for Osteopenia

**DOI:** 10.3390/nu15163515

**Published:** 2023-08-09

**Authors:** Serafina Perrone, Chiara Caporilli, Federica Grassi, Mandy Ferrocino, Eleonora Biagi, Valentina Dell’Orto, Virginia Beretta, Chiara Petrolini, Lucia Gambini, Maria Elisabeth Street, Andrea Dall’Asta, Tullio Ghi, Susanna Esposito

**Affiliations:** 1Neonatology Unit, Pietro Barilla Children’s Hospital, University of Parma, Via Gramsci 14, 43126 Parma, Italy; valentinagiovanna.dellorto@unipr.it (V.D.); virginia.beretta@unipr.it (V.B.); cpetrolini@ao.pr.it (C.P.); lgambini@ao.pr.it (L.G.); 2Pediatric Clinic, Pietro Barilla Children’s Hospital, University of Parma, Via Gramsci 14, 43126 Parma, Italy; caporillichiarasp@gmail.com (C.C.); fedigrassi@hotmail.it (F.G.); mandy.ferrocino@unipr.it (M.F.); ele.biagi97@gmail.com (E.B.); mariaelisabeth.street@unipr.it (M.E.S.); susannamariaroberta.esposito@unipr.it (S.E.); 3Obstetric and Gynecology Unit, University Hospital of Parma, University of Parma, Via Gramsci 14, 43126 Parma, Italy; andrea.dallasta@unipr.it (A.D.); tullio.ghi@unipr.it (T.G.)

**Keywords:** newborn infant, bone mineral density, ultrasounds, oxidative stress, endocrine disruptors

## Abstract

Bone health starts with maternal health and nutrition, which influences bone mass and density already in utero. The mechanisms underlying the effect of the intrauterine environment on bone health are partly unknown but certainly include the ‘foetal programming’ of oxidative stress and endocrine systems, which influence later skeletal growth and development. With this narrative review, we describe the current evidence for identifying patients with risk factors for developing osteopenia, today’s management of these populations, and screening and prevention programs based on gestational age, weight, and morbidity. Challenges for bone health prevention include the need for new technologies that are specific and applicable to pregnant women, the foetus, and, later, the newborn. Radiofrequency ultrasound spectrometry (REMS) has proven to be a useful tool in the assessment of bone mineral density (BMD) in pregnant women. Few studies have reported that transmission ultrasound can also be used to assess BMD in newborns. The advantages of this technology in the foetus and newborn are the absence of ionising radiation, ease of use, and, above all, the possibility of performing longitudinal studies from intrauterine to extrauterine life. The use of these technologies already in the intrauterine period could help prevent associated diseases, such as osteoporosis and osteopenia, which are characterised by a reduction in bone mass and degeneration of bone structure and lead to an increased risk of fractures in adulthood with considerable social repercussions for the related direct and indirect costs.

## 1. Introduction

Bone health begins with maternal health and nutrition, which influence skeletal mass and bone density in the foetus [1]. Significant variability in terms of bone mass has been demonstrated in the general population, which appears to originate as early as the foetal and early post-natal period [2].

Although the weight of subjects at one year of age has been most studied in the literature as it correlates with peak bone mass in adulthood, a recent systematic review has shown a strong correlation of adult bone mass with birth weight as well, suggesting an influence of environmental stimuli received in the critical period of skeletal development since intrauterine life [3,4].

It has recently been discovered that mechanisms mediated by epigenetics—among them, microRNAs—are able to control gene expression at the post-transcriptional level, providing epigenetic modifications and directing the growth plate.

Metabolic bone disease (MBD) is defined as a decrease in bone mineral content compared to the expected level for a child of a certain weight or gestational age, identifiable through alterations in laboratory and radiographic examinations [5].

Populations most at risk of developing MBD are preterm births, infants with low birth weight—particularly very low birth weight (VLBW) and extremely low birth weight (ELBW)—infants with intrauterine growth restriction (IUGR), infants with comorbidities typically associated with prematurity (like sepsis, cholestasis, bronchopulmonary dysplasia (BPD), necrotizing enterocolitis (NEC)), infants requiring long periods of total parenteral nutrition (TPN), infants born to mothers with pregnancy-associated diseases (preeclampsia, chorioamnionitis, gestational diabetes), and infants born to vitamin D-deficient mothers [5]. Infants with these characteristics should be carefully and adequately screened, including laboratory, urinary, and instrumental tests, detect any signs of MBD early and undertake appropriate mineral supplementation as soon as possible on the basis of laboratory data.

Most commonly, MBD occurs between 6 and 16 weeks after birth [6]; it manifests due to inadequate calcium and phosphorous reserves and is exacerbated by inadequate mineral intake and the high skeletal growth rate, which normally occurs in the first few weeks of life [6,7].

MBD is most common in newborns <28 weeks of gestation; the incidence of MBD in preterm VLBW was 32% and for ELBW 54%, with a peak at 4–8 weeks of postnatal age [8].

Effective prevention, early identification of risk factors, the use of unified diagnostic algorithms, and non-invasive technologies can help improve bone health and decrease the incidence of MBD and associated morbidities that represent a substantial cost to healthcare.

Challenges for bone health prevention include the need for new technologies that are specific and applicable to pregnant women, the foetus, and, later, the newborn. This educational review aims to describe the current evidence for early identification and management of populations at high risk for osteopenia and the implementation of screening and prevention programmes based on gestational age, weight, and morbidity.

## 2. Foetal Skeletal Development

Bone tissue is made up of three types of cells: osteoblasts, osteocytes, and osteoclasts. These originate from a matrix consisting mainly of collagen fibres, glycoproteins that give shape and elasticity to the bone, and mineral salts, mainly calcium and phosphorus, which give rigidity and hardness. Bone tissue is also metabolically active and extremely dynamic tissue due to its continuous remodelling throughout life [9].

Skeletal development starts in the first weeks of gestation and is influenced by genetic, endocrine, and environmental factors [10]. Proliferation, differentiation of cartilage precursors, and ossification processes are finely regulated by hormones such as parathyroid hormone (PTH), cytokines, and vitamins (A, D, C, and K) [11].

Another key element for good skeletal development is vascular support that allows sufficient mineral supply. In fact, conditions such as pre-eclampsia, chorioamniosite, and IUGR, reflecting chronic placental injury, are associated with an increased risk of MBD [12].

In addition, several environmental factors, such as drug exposure, toxic substances, and maternal nutritional conditions [13], are considered epigenetic factors influencing foetal bone mass and density, as well as bone quantity and quality in adults [14].

Embryologically, bone tissue is derived from the mesoderm, a leaflet that generates all connective tissues during foetal life. The first skeletal structure to be generated is the notochord, which appears in the midline of the embryonic disc around the 15th day of development, originating from a primitive dimple located at the elevation of the ectoderm, known as Hensen’s node, at the end of the primitive line [15].

Specifically, the craniofacial skeleton originates from neural crest cells, the axial skeleton is formed from paraxial mesodermal cells, and the appendicular skeleton is formed from lateral plate mesodermal cells [16].

Bone formation can occur through two models of ossification: intramembranous or endochondral [17].

Intramembranous or direct ossification occurs when bone tissue replaces cartilage tissue. Mesenchymal cells proliferate and condense into compact nodules. Some of these cells form capillaries; others are induced by bone morphogenetic protein (BMP) to express Runx2 and differentiate into osteoblasts that, in turn, begin to produce an osteoid matrix that generates bone tips after the mineralization process [18]. Intramembranous ossification represents the usual means of the skull, mandible, and clavicle flat bone development. The membranous bones of the braincase and the face form between the 9th and 12th week [16].

Endochondral or indirect ossification requires the formation of cartilaginous tissue from aggregate mesenchymal cells and subsequent replacement of cartilage with bone. Endochondral ossification gives rise to most of the bones in the human body [16,17], which are also those with a higher risk of rupture in later age [18].

Indirect ossification may be divided into five successive phases. During this process, chondroblasts must generate, proliferate, experience hypertrophy, die, and be replaced by osteoblasts. Sequential changes in chondroblast behaviour are rigidly controlled by systemic and paracrine factors that, by binding receptors, activate intracellular signalling pathways and determine the activation of specific transcription factors, such as paired box l (PAXl) and scleraxis, responsible for activating cartilage-specific genes. The offspring cells that deposit the cartilage are located at the extremities of the long bone [19,20,21]. The drafts of the upper limbs appear first (24th day), followed by those of the lower limbs (28th day). The limb drafts initially consist of a mesenchymal nucleus derived from the somatopleura of the lateral mesoderm. After five weeks, the coccygeal vertebrae are completed in the embryo [15].

At the end of the 5th week, well-formed cartilage sketches represent all future major bones of the limb skeleton [18]. This cartilage model is subsequently replaced by newly formed bone [17]. The first changes in the cartilaginous contour of a long bone occur in the central part of the diaphysis around the 7th week of embryonic life when the developing bone is invaded by multiple blood vessels—one of which becomes the nourishing artery of the bone. The foetal cartilaginous skeleton and the formation of primary ossification centres in vertebrae and long bones complete their development in the first trimester [22]. The first bone in which the primary ossification centre appears is the femur. The last bone is the hyoid bone, where the ossification centre appears at 36 weeks. Some of the smaller bones of the carpus and tarsus only begin to ossify during early childhood. This ossification process proceeds from the primary ossification centre to the epiphyses. At birth, the diaphyses are completely ossified, while the ends of the bones, called epiphyses, are still cartilaginous. Consequently, after birth, secondary ossification centres appear in all bones at the level of the epiphyses. However, between the epiphysis and the growing end of the diaphysis, a layer of cartilage called epiphyseal cartilage persists. In this layer, a continuous proliferation of chondrocytes, followed by differentiation and replacement of cartilage tissue by bone tissue, allows the diaphysis to elongate. The epiphyseal plate ossifies completely at around the age of twenty when the growth of the body is complete [23].

## 3. The Role of Epigenetics in the Regulation of Placenta and Foetal Bone Development

Each stage of skeletal development is regulated by numerous transcription factors (e.g., SOX9, RUNX2), growth factors (e.g., FGF, IGF, VEGF, BMP, and others of the TGF-β family), and other signalling molecules (e.g., Wnt/β-catenin, Hedgehog, PTHrP) [24].

It has been hypothesized that various epigenetic changes in the placenta, which are responsible for the transfer of nutrients to the foetus, also play a role in the abnormal development of the foetal skeleton and, thus, osteoporosis in adulthood. This could be hypothesized with the discovery of the association between gene dysregulation and osteopenia [13,24].

Recent studies have shown, in particular, that enzymes capable of modifying histone structure or controlling DNA methylation and non-coding microRNAs also play a fundamental role in skeletal development as epigenetic factors capable of acting at transcriptional and post-transcriptional levels on gene expression [25].

Overexpression of some microRNA groups (e.g., miR-10b, miR-19a-3p, miR-29a, miR26b) in vitro induces osteoblast differentiation while inhibition of others (e.g., miR-9-5p, miR-16-2-3p) suppresses osteogenesis [25]. In preclinical studies, overexpression of osteo-inducible microRNAs seems to be able to induce the formation of mineralised extracellular matrix and the expression of genes related to osteoblastic function in human or murine mesenchymal progenitor cells through different epigenetic mechanisms: the inhibition of epigenetic regulators, such as HDAC4 or HMGA2, the inhibition of the Wnt-mediated signalling cascade (e.g., DKK1, GSK3β), the modulation of negative regulators of RUNX2 (e.g., SMAD6 or SMURF2), and the regulation of Rorβ [26,27,28].

WNT2 genes encode a protein involved in cell signalling pathways, plays an important role in mouse placental development, and is highly expressed in the human placenta [29]. Ferreira et al. showed an association between WNT2 promoter methylation in the human placenta and low birth weight [29].

Tenta et al. identified a role in the induction of osteoclastogenesis by the nuclear factor-kappaB activating receptor (RANKL) and the RANK decoy osteoprotegerin (OPG) receptor, revealing a significant overregulation of the OPG/RANKL ratio in SGA infants, highlighting their role in bone turnover in compensating for intrauterine growth retardation [10].

Studies have shown polymorphisms in PTH pathway genes (PTH, PTH-like hormone (PTHLH), and PTH1 receptor (PTHR1)), suggesting that PTH-related genes are strong candidates for the genetic regulation of bone development and bone loss. There is evidence that they may be associated with fracture risk or BMD [30,31,32].

Other risk factors for MBD in preterm infants are male gender and polymorphisms of certain genes (vitamin D receptor, oestrogen receptor, and collagen alpha 1 genes) [33].

Data show that vitamin D and vitamin D receptor play a role in bone growth, impacting cell proliferation and differentiation, and, in particular, calcium homeostasis, all of which are critical processes for bone health [34].

COL1A1 Sp1 plays a role in foetal bone development. COL1A1 Sp1 for type 1 collagen is the gene transcript for a major protein of bone. Osteoporotic fractures and BMD have been found to be associated with COL1A1 Sp1 polymorphisms [35].

Other genes such as Sox9, Runx2, Osterix, ALP, osteocalcin, and bone sialoprotein (Bsp) play a role in osteoblast differentiation and, thus, bone formation [36,37].

## 4. Bone Mineralization

Bone mineralization is an indispensable process for proper skeletal formation [38]. Maintaining adequate mineral content is critical for effective bone mineralization [39]. Minerals are absorbed by the intestines and recovered or excreted by the kidneys as needed. Dietary intake is important to ensure mineral homeostasis and prevent mineral bone diseases, as well as balancing the actions of different regulators of biomineralization, including parathyroid hormone, calcitonin, vitamin D, vitamin K, fibroblast growth factor 23 (FGF23), and phosphatase enzymes. Indeed, several factors affect bone mineralization, including adequate mineral intake, the balance between mineralization activators and inhibitors, and the presence of collagen fibrils [38]. The body also uses the skeleton as a source of minerals in case of deficiency. Bone plays an important role as a metabolic store that regulates intracellular and extracellular levels of minerals, particularly calcium and phosphate [38].

About 70 per cent of the weight of bone is made up of minerals, the remaining 30 per cent of organic material. The mineral part is mostly represented by hydroxyapatite crystals, highly organized calcium and phosphate structures, and other ions such as sodium, magnesium, fluoride, and strontium. Whereas the inorganic part consists of collagen fibres, glycoproteins, and proteoglycans [40].

Bone mineralization is a two-step biological process regulated by several factors: primary and secondary mineralization. In the primary mineralization, deposition of amorphous calcium–phosphate salts occurs and in the secondary mineralization, progressive mineral maturation to form hydroxyapatite [41].

Bone mineralization begins from small extracellular matrix vesicles secreted by chondrocytes and osteoblasts [42].

The inflow of calcium and inorganic phosphate ions into these small vesicles is regulated by membrane transporters and enzymes involved in mineralization (i.e., tissue nonspecific alkaline phosphatase (TNAP), ectonucleotide pyrophosphatase (ENPP), and ankylosis (ANK)) [43,44,45].

In primary mineralization, the accumulation of phosphate and calcium within the matrix vesicles results in the nucleation and progressive accretion of hydroxyapatite crystals, which gradually form mineralized nodules [46].

These nodules, also called calcifying nodules, make contact with collagen fibres resulting in collagen mineralization from the contact points to the periphery [47].

During secondary mineralization, bone mineral density increases progressively due to mineral transport processes involving osteocytes. Osteocytes have a lacunar osteocytic canalicular system that enables them to transport minerals and regulate bone metabolism [48].

In this way, osteocytes and osteoblasts cooperate to maintain an adequate state of bone mineralization [49].

Calcium and inorganic phosphate serum levels are two important determinants of bone mineralization [50].

Furthermore, 80% of calcium and phosphorous transfer occurs between 25 and 40 weeks of gestation, and the maximum amount is reached at 34 weeks [51,52,53].

In the foetus, the daily accumulation of calcium at the 24th week is 60 mg, while between the 35th and the 40th week of gestation, it comes to be equal to 300–350 mg. The total average spawning of calcium and phosphorus is 100–150 mg/kg/day and 50–65 mg/kg/day, respectively [54].

The extracellular calcium level is regulated by PTH, the secretion of which depends on the blood calcium concentration detected by calcium-sensitive receptors (CaSR) on parathyroid cells. PTH secretion activates mechanisms to increase serum calcium levels, including bone resorption by osteoclasts [40].

Blood phosphate levels depend on the absorption of dietary phosphate by the intestines, on the release of phosphate from the bone, and especially on its recovery along the nephrons. Excretion and reabsorption of phosphate filtered by the kidney occur in the proximal tubule via specific transporters, according to the needs of the body, and are the main mechanisms for regulating blood phosphate content [55].

The activity of these transporters is modulated by PTH and FGF23, a hormone produced by osteocytes, resulting in increased phosphate excretion and reduced blood phosphate levels [56].

FGF23 and PTH also affect vitamin D metabolism and bone metabolism with opposite actions. PTH increases serum calcium and phosphate levels by stimulating vitamin D activation and promoting intestinal absorption of minerals. Instead, FGF23 inhibits the formation of calcitriol, the active form of vitamin D [38].

FGF23 also inhibits bone mineralization by suppressing the synthesis and secretion of PTH and especially by promoting the formation of pyrophosphate (PPi), one of the most potent inhibitors of mineralization [57].

PPi hinders the aggregation of calcium and phosphate to prevent the formation and growth of hydroxyapatite crystals [58]. It is hydrolyzed by the enzyme alkaline phosphatase (ALPL); this process reduces PPi levels that inhibit bone mineralization and, at the same time, increases the available amount of phosphate [50].

The indispensable function of ALPL is evidenced by the fact that newborns with mutations in the ALPL gene exhibit severe osteomalacia, foetal/perinatal lethality [59].

Another important regulator of calcium and phosphate homeostasis is vitamin D. The final step of the transformation of the inactive form of vitamin D into its active form, calcitriol, is stimulated by various factors, including PTH, insulin-like growth factor 1, a reduced intake of calcium and phosphate, and low extracellular levels of these minerals [38].

Vitamin D promotes the maintenance of an adequate plasma content of calcium and phosphate, stimulating the absorption of minerals from the intestine and their mobilization from the bone [60].

The achievement of supersaturated extracellular concentrations of calcium and phosphate allows the mineralization of hypertrophic cartilage and bone. Calcitriol also promotes mineralization by stimulating osteoblast differentiation, the growth of the matrix vesicles from which the process originates, and the production of matrix non-collagenous proteins involved in the regulation of mineralization [61].

A recent review shows the important role of vitamin K in bone metabolism and mineralization. Vitamin K, due to its anabolic properties, promotes Y-glutamyl carboxylation and osteoblast differentiation, regulates extracellular matrix mineralization through the activation of bone-associated vitamin K-dependent proteins, and inhibits osteoclastogenesis. All effects stimulate bone formation and mineralization [9].

## 5. Factors Affecting Bone Mineralization 

Many factors, acting in pre and postnatal period, can affect bone mineralization (Figure 1)

### 5.1. Maternal and Fetal Factors

#### 5.1.1. Vitamin D

The association between MBD and vitamin D levels in gestating women is still debated: studies show no conclusive results. Some studies have found no correlation between maternal vitamin D levels and bone parameters in the neonatal period [62,63].

Other studies show a negative association with offspring bone mass density [64] or a positive association [65,66].

Other studies suggest that vitamin D supplementation should be given to pregnant women in order to ensure sufficient vitamin D levels for the newborn, as neonatal concentrations of 25-hydroxyvitamin D (25(OH)D) are approximately 80% of maternal levels [67,68].

In addition, clinical studies have shown that vitamin D supplementation in pregnant women reduces the risk of MBD because it reduces the likelihood of preeclampsia and gestational diabetes, which are risk factors for MBD [69,70].

#### 5.1.2. GH/IGF-1 Axis

The growth hormone/insulin-like growth factor 1 (GH/IGF-1) axis has been considered crucial in the acquisition of bone mass. Foetal growth restriction is likely to have a negative effect on this axis [71].

Insulin-like growth factor (IGF-1) promotes mitosis of differentiated chondrocytes, both in the foetal period and in infancy, and is one of the most important endocrine factors. Insufficient secretion is accompanied by reduced growth levels, and this may be the reason for lower bone mass in children born with a lower birth weight [72].

#### 5.1.3. Cortisol

In a meta-analysis, an association, although weak, was also demonstrated between infants born with a lower birth weight and serum cortisol levels: this category of infants had elevated cortisol levels [73].

Endogenous cortisol, in fact, inhibits the function of osteoblasts, so high levels may negatively affect bone mass [74,75].

#### 5.1.4. Leptin

Leptin is a cytokine hormone secreted by adipocytes; among its many functions, it also has the ability to link changes in body composition with bone formation and resorption [76].

#### 5.1.5. Oxidative Stress and Endocrine Disruptors

Oxidative stress (OS) and endocrine disruptors (EI) in pregnant women could potentially affect maternal and foetal bone health through foetal programming. Oxidative stress can cause intrauterine growth retardation in the most common pregnancy conditions, such as maternal diabetes, pre-eclampsia, and chorioamniosis, generating an intrauterine environment potentially adverse to foetal development [77].

In fact, during pregnancy, there is an excess production of free radicals in relation to the body’s antioxidant capacity. Sources of free radicals come from the mitochondrial respiratory chain, the induction of nitric oxide synthases (NO synthases) and nicotinamide adenine dinucleotide phosphate oxidase (NADPH oxidase), and from free metals such as iron, copper, and manganese [77].

Free radicals can, through multiple epigenetic mechanisms, modulate gene expression at critical periods of foetal development directly or indirectly through the molecules oxidised by the radicals themselves [77].

Longini et al. demonstrated that isoprostanes in amniotic fluid are a reliable marker of lipid peroxidation: free radicals, by breaking the peptide bonds of the collagen amino acid chain and peroxidising the polyunsaturated fatty acids of the cellular phospholipid bilayer, damage the structure of the chorio-amniotic membranes, predisposing women to premature rupture of the membranes and thus to preterm delivery [78].

Endocrine disruptors (EDCs) appear to be able to act on the placenta by modifying its morphology and function with mechanisms that are still largely unknown but are probably linked to signalling pathways associated with insulin, glucocorticoids, oestrogen, and thyroid hormones, epigenetics, and inflammation [79,80,81,82].

The literature on the outcomes of exposure to endocrine disruptors during foetal life is of great interest, although still limited, and has focused on possible repercussions on foetal growth and long-term cognitive–behavioural adverse events [83].

Recently, Dirkers et al., in a preclinical study in a mouse model, investigated how bisphenol A exposure during pregnancy leads to a weakening of the trabecular microarchitecture and cortical geometry in exposed offspring. The study group, compared to the control group, had a statistically significant reduction in trabecular bone volume and cortical thickness along with increased trabecular space, and, ultimately, a weakening of bone [84].

There are no data to date related to early exposure to EDCs and changes in bone quality and composition in humans. Some of these changes could be exercised by epigenetic changes and, among these, by changes in the miRNA network. [13]. miRNAs are endogenous small non-coding RNAs that act as transcriptional and post-transcriptional regulators. Multiple changes in miRNA abundance can occur, where simultaneously up- and down-regulated miRNAs can target the same gene with a range of predicted effects and, vice versa, a single miRNA can regulate several target genes. miRNAs play a role in the regulation of skeletal development, with effects at different bone developmental stages and in different bone cell types [85,86]. A study in newborns and children showed different miRNA expression levels dependent on different developmental stages [87]. In particular, 36 miRNAs increased from birth to mid-childhood. The molecular mechanisms by which miRNAs exert their regulatory role in longitudinal bone growth are involved with the regulation of cell growth, particularly of chondrocytes [88]. 

Chondrocytes are involved in the growth of the skeleton: endochondral bone formation occurs in the growth plate, which is a thin layer of cartilage located in the metaphysis of long bones. The growth plate shows a high degree of spatial regulation and is histologically formed by three distinct zones: the resting zone, which contains progenitor chondrocytes; the proliferative zone, characterized by chondrocytes that proliferate unidirectionally to form columnar cell clones and produce specific extracellular matrix proteins (e.g., type II collagen and aggrecan); and the hypertrophic zone, in which mature chondrocytes exit the cell cycle, go through hypertrophic differentiation, and express type X collagen. The cartilage matrix becomes mineralized, and terminally differentiated chondrocytes undergo apoptosis [89].

The expression pattern of miRNAs may be involved in the control of proliferative and differentiative mechanisms which regulate the cell fate of the specific growth plate zones. Moreover, the distinct patterns in the growth plate are influenced by parathyroid hormone-related protein (PTHrP) concentration gradients across the zones, suggesting a role for miRNAs in the mechanism of action of PTHrP for the control of the growth plate cell [90], Figure 2.

Furthermore, miRNAs are critical for the regulation of hypothalamus function and pituitary development, and thus for GH secretion and action also. It has been demonstrated that three specific miRNAs that regulate the genes involved with bone growth are directly regulated by growth hormones and are of great interest with respect to longitudinal growth, namely miR-199a-5p, miR-335-5p, and miR-494-3p. miR-199a-5p and miR-335-5p play a role in bone formation and osteoblast differentiation in vitro. In particular, miR-199a-5p is involved in osteoblast differentiation, and the overexpression of miR-335-5p has been reported to promote bone formation and regeneration in a transgenic mouse model [91].

miR-494-3p has not been studied yet in the context of bone or growth plate development; however, it has been reported to promote the PI3K/AKT pathway [92], which is known to control hypertrophic chondrocyte differentiation and to be involved in endochondral bone growth and osteoblast differentiation [93,94].

Data suggest that in the near future, nutrients or drugs targeting this pathway or gene therapy may play a role in patients who can be defined as high risk to prevent the development of osteopenia and osteoporosis later in life.

#### 5.1.6. IUGR

Wang’s meta-analysis pointed out that IUGR is an independent risk factor for MBD [95].

Olmos-Ortiz et al. found a correlation between IUGR and maternal vitamin D3 deficiency; the latter leads to defective implantation of the placenta [96].

This event leads to chronic malfunction of the placenta with a blockage of phosphorus transport and poor intrauterine bone calcification [97].

### 5.2. Neonatal Factors

#### 5.2.1. Preterm Birth

Wang et al. highlight how gestational age < 32 weeks is a risk factor for MBD [95].

This is mainly related to two reasons: (1) bone mineralization occurs predominantly in the third trimester, and premature birth causes a lack of foetal mineral reserves; (2) premature babies hospitalised for various reasons (low gestational age or need for assisted ventilation) are immobile, so there may be a risk of bone mineralisation defects [98,99]; furthermore, immobility could be due to inactivity due to metabolic, neuromuscular, or systemic diseases [100].

If premature birth occurs between 24 and 40 weeks of gestational age, newborns miss the optimal phase for obtaining maximum calcium stores and phosphorus and often do not have adequate supplies of these minerals, so they are particularly sensitive to changes in the postnatal period and, moreover, due to comorbidities associated with prematurity, they are exposed to risk factors for reduced bone mineralization [101,102].

In preterm infants, the bone mineral content is insufficient for normal bone growth; this may be evidenced by alterations in imaging and electrolytes and abnormal levels of certain enzymes in the blood [5].

#### 5.2.2. Low Birth Weight

Low-birth-weight infants can be related to serious metabolic abnormalities that occur in adult life, such as metabolic syndrome and osteoporosis [103]. Low birth weight has been shown to be a determinant of low bone mass [104].

As many studies have shown, birth weight < 1000 g is also a risk factor for MBD; indeed, as birth weight increases, the risk of MBD decreases. There are mainly two reasons for this: (1) premature birth, as explained above [18]; (2) alterations in placental function resulting in impaired nutrient and mineral transfer [51].

A systematic review and meta-analysis correlates birth weight and adult bone mineral content, demonstrating a positive association. This meta-analysis shows that the bone mineral content of the hip increases as birth weight increases [2].

However, there is some evidence that later growth is a more important factor than birth weight [105,106].

#### 5.2.3. Preterm Morbidities

The common neonatal morbidities, especially in the preterm infant (sepsis, chronic pulmonary acidosis, necrotising enterocolitis, cholestatic jaundice), and long-term treatment with drugs such as diuretics and glucocorticoids, which preterm infants often need, can impair bone remodelling. The proliferative activity of osteoblasts is reduced and the demolition activity of osteoclasts is promoted, decreasing calcium absorption and increasing urinary calcium loss [107,108,109].

Children with sepsis have a higher risk of MBD [110,111]. This correlation is due to the interaction between the immune system and bone metabolism [112]. Exposure to lipopolysaccharides can cause loss of bone density [113], as it leads to the activation of B and T cells that regulate bone resorption [112].

In addition, the treatment of sepsis and the disease itself, associated with the physical decay of the newborn, can prolong the use of parenteral nutrition with an increased risk of MBD [111].

The cholestasis leads to decreased vitamin D absorption. Premature infants, especially those with a gestational age < 32 weeks, usually have low serum levels of 25-hydroxyvitamin D [114].

Lithocholic acid increases cholestasis and acts as a vitamin D analogue by reducing its absorption [115].

Furthermore, bilirubin and bile acid increase and inhibit the function of osteoblasts, with negative effects on the mineralization process [116].

Lee et al., in their retrospective case-control study, examined the medical history of 55 infants admitted to the Neonatal Intensive Care Unit at Severance Children’s Hospital. Their study demonstrated a positive association among ELBW infants with parenteral nutrition-associated cholestasis, BPD, and poor bone mineralization [107].

#### 5.2.4. Iatrogenic Factors

Iatrogenic risk factors can be identified, such as taking drugs that can alter mineral levels and long-term parenteral nutrition [6].

The cornerstones of chronic lung disease therapy are diuretics and steroids; by their mechanism of action, these drugs result in calcium mobilization from bone, exacerbating MBD—they reduce osteoblast proliferation, stimulate osteoclast activation, decrease calcium absorption, and increase renal excretion [117].

Calcium deficiency can lead to metabolic changes associated with secondary hyperparathyroidism. Increased PTH levels lead to loss of phosphate in the urine and, consequently, hypophosphatemia. Increased PTH causes an elevation of serum calcium due to increased bone reabsorption and renal and intestinal absorption of calcium. Isolated plasma calcium levels may, therefore, not be a useful screening indicator for children at risk of MBD [5].

Long-term parenteral nutrition is often necessary for preterm infants who fail to be enteral fed in the postpartum period and/or fail to achieve full enteral nutrition in the short term [118].

If TPN is required for a period longer than 4 weeks, low calcium and phosphate intake is common [101].

Patients on parenteral nutrition struggle to reach normal serum mineral levels—mineral intake fails to reach even 50% of the levels reached during foetal life [119].

A lack of mineral formulations, poor solubility of minerals, and pH interferes with parenteral nutrition formulations by not allowing sufficient mineral supply. The solubility of calcium and phosphate is influenced by environmental conditions, such as temperature and the content of amino acids, glucose, and lipids, as well as the pH of the solution [118,120].

Therefore, the bone deposition of minerals, such as calcium and phosphorus, in the first weeks of life of preterm infants fails to meet the levels required for proper bone growth, comparable to intrauterine growth [101]. Additionally, studies show aluminium contamination during parenteral nutrition, which can lead to MBD [121], as there is excessive deposition of aluminium on the surface of bone mineralization, which hinders bone formation, preventing osteoblasts from functioning [122].

## 6. Screening and Monitoring

The importance of maximising bone health is now recognised by doctors, so many screening tests have been devised, but the optimal one has not yet been identified [12,123].

Risk factors guide us in choosing what to screen for [12,118].

In preterm infants, the bone mineral content is not sufficient for normal bone growth; this can be evidenced by electrolyte alterations and abnormal levels of some hormones in the blood and imaging [5].

To improve linear growth in preterm infants with risk factors for MBD, regular follow-up and monitoring are necessary. The objectives are to maintain blood calcium and blood phosphorus within normal limits and avoid excessive urinary calcium excretion [95].

Kolisambeevi et al. carried out a prospective study with the aim of reducing the incidence of MBD in VLBW infants with gestational age ≤ 30 weeks [124].

According to this study, monitoring of phosphorus is necessary from the first days of life and supplementation, intravenous and/or oral, from the 4th day of life. When reaching 40 mL/kg/day of enteral human milk, breast milk fortifier is gradually introduced. The MBD rate is 2.8% in newborns with a gestational age between 28 and 30 weeks and del 69.2% in those with gestational age ≤ 26 weeks. With the results obtained, the authors promote the mandatory early use of phosphorus supplements [124].

For early diagnosis of mineral deficiency, measurement of serum biochemical markers is essential. The most commonly used blood biochemical indicators are serum calcium, serum phosphorus, ALP, PTH, and 25(OH)D [95].

Calcium: Hypocalcemia (corrected values for albuminemia) is suggestive if <8.5 mg/dL. Serum calcium levels are regulated by calcitonin and PTH. When circulating calcium decreases, the body mobilizes calcium underneath the bone, stimulating PTH to maintain adequate serum calcium; in addition, serum calcium levels can also be affected by hypophosphatemia [102].

Blood calcium only decreases when bone calcium stores are depleted, the late stage of MBD. Therefore, the diagnosis of MBD in the early stage with blood calcium does not make sense [95].

Phosphate: phosphoremia is a good indicator for assessing bone phosphorus reserves; a persistent decrease in serum phosphorus indicates inadequate supply, which increases the risk of osteoporosis. When phosphorus levels remain persistently low in the blood, bone resorption increases, urinary calcium excretion continues to increase, and this leads to calcium depletion [95].

Phosphate levels are suggestive if <5.5 mg/dL; it is the first marker of poor bone mineral metabolism and appears at 7–14 days of age. In newborns exclusively breastfed, serum phosphorus < 3.6 mg/dL suggests a severe deficiency of the mineral content, thus a higher risk of developing MBD [125].

During the first months of life in preterm infants, especially those with breathing problems and gastrointestinal complications or who have taken methylxanthines, diuretics, or corticosteroids, serum phosphorus should be monitored carefully [51].

Hypophosphatemia also stimulates the renal tubular synthesis of vitamin D with consequent promotion of intestinal absorption of calcium. Thus, phosphate deficiency may be responsible for hypercalcemia, hypercalciuria, and nephrocalcinosis [5].

Alkaline Phosphatase (ALP): ALP, a marker of bone turnover, increases physiologically during the first 3 weeks of life with a peak at 6–12 weeks of age [126].

ALP levels > 500 IU/L indicate impaired bone homeostasis. If the values are 700 IU/L, they indicate a demineralization, even in asymptomatic patients; in fact, increased levels of alkaline phosphatase in the blood may precede the onset of clinical symptoms [95,127]. Some studies for VLBW infants suggest a wrist and/or knee radiograph when at least two ALP values > 800 IU/L are recorded after 1 week [128].

Viswanathan et al. demonstrated that in ELBW infants with <30 weeks of gestation, the finding of ALP > 500 IU/L is associated with MBD [117].

ALP > 1000 IU/l can be considered a marker of rickets [129]. Using ALP in conjunction with blood phosphorus, specificity increases. ALP > 900 IU/L with serum phosphorus < 5.6 mg/dL (<1.8 mmol/L) has a sensitivity of 100% and 70% specificity [130]. The measurement of serum phosphate and ALP has been recommended weekly or biweekly [131].

Parathyroid hormone (PTH): The plasma concentration of calcium ions regulates the secretion of PTH. Studies show that the concentration of PTH can be used as a tool, along with other parameters, to guide the screening and monitoring of MBD [132]. In particular, serum PTH levels can be a useful marker to identify ELBW infants at risk of MBD: PTH > 100 pg/mL may suggest ELBW infants at risk of MBD [133]. Studies suggest that serum PTH levels could predict a decrease in bone mineral contained in preterm infants who have reached full-term age [134].

PTH is a marker of secondary hyperparathyroidism and, in combination with tubular reabsorption of phosphorus (TRP), can help to distinguish the causes of hypophosphatemia. A low TRP with high PTH levels could indicate a calcium deficiency. A high TRP with low PTH levels or normal would instead indicate a lack of phosphorus [5].

When phosphorous supplements alone are used for low-birth-weight infants, there is a stimulation of secondary hyperparathyroidism that may increase the risk of MBD [51].

Elevated serum PTH levels were found in over 80% of ELBW infants with signs of osteopenia [133].

Vitamin D (25(OH)D): In the case of MBD, serum 25(OH)D may be normal, decreased, or increased, so it is not used as a screening marker [95].

Several studies have documented an association between low 25(OH)D levels and low gestational age at delivery, so premature infants represent a vulnerable population with regard to their vitamin D status [114].

Careful monitoring of vitamin D levels is not necessary if there are no contraindications for routine vitamin D supplementation, as it is expensive monitoring [67]. As supported in the literature, 25(OH)D levels > 50 nmol/L prevent rickets in infants and children [135].

Recent literature, also on neonatal populations, demonstrates normalization of PTH to levels > 75 nmol/L when adequate serum levels of 25(OH)D are present [136].

Serum osteocalcin (OC): Serum osteocalcin is a bone matrix protein which increases during high bone turnover. However, there is no evidence that increased serum OC levels correlate with bone mineral content in the first four months of life [137]. Czech-Kowalska et al. suggest that OC and urinary phosphate excretion could be markers of poor bone mineralisation at 3 months of corrected age [134].

Urinary markers: Urinary biochemical indicators include urinary phosphorus, urinary calcium, urinary phosphorus/creatinine, urinary calcium/creatinine, and TRP [12]. The formula for calculating the TRP is as follows: [1 − (urinary phosphorus/urinary creatinine × serum creatinine/serum phosphorus)] × 100. The normal value of TRP is 78–91%; it is a marker of insufficient phosphate integration for values > 95% [138]. Babies born < 28 weeks of gestation have a lower phosphate value than other preterms, with high urinary phosphate excretion, even in the presence of low serum phosphate levels. Renal reabsorption of phosphate increases consequently to the rise in PTH levels, caused, in turn, by hyposfatemia. There are no clear monitoring guidelines, although many clinicians agree to monitor laboratory tests every 1–2 weeks. Abrams et al. recommend performing blood phosphorus and ALP measurements weekly or biweekly [118]. Land et al. recommend measurements of serum calcium and phosphorus, accompanied by calciuria and phosphaturia, weekly in premature infants less than three weeks old and biweekly in those more than 3 weeks old [139]. Harrison et al. recommend weekly measurement of serum levels such as serum calcium and phosphorus and also ALPs and TRPs [12].

X-rays are only useful in severe MBD with significant signs of osteoporosis or bone fractures; osteoporosis with <20–40% bone loss may not be evident with this method, so they are not suitable for early diagnosis. Alterations detectable by X-rays are demineralization or “osteopenia”, rachitic changes, osteoporosis of the ends of the long bones, epiphyseal changes (cupped or burr), widening of the ends of the ribs, subperiosteal new bone formation, or fractures [12,95].

Koo et al. describe the radiological alterations: grade (1) bone rarefaction; grade (2) bone rarefaction e metaphyseal changes and subperiosteal bone formations; grade (3) spontaneous fractures [140].

Some fractures can be acute, manifesting in pain or inability to break movement, but more commonly, fractures are asymptomatic [141]. The American Academy of Pediatrics recommends radiographs every 5–6 weeks until mineralization improves [118].

The gold standard for the identification of osteoporosis is dual-energy X-ray absorptiometry (DEXA) [142], suitable for preterm infants [143]. DEXA calculates BMD [142,144], that is, the content of bone calcium in grams of hydroxyapatite per square centimetre, by using a low dose of ionizing radiation (range of 1–13 μSv) [145,146].

In clinical practice, however, it is difficult to use this instrument in preterm and term infants for various reasons: the instrumental dimensions, the time taken to motion artifacts; its use for MBD screening is not suitable for routine screening, as it is difficult to perform [95,102]

Quantitative Ultra Sound (QUS) measures bone mineral content and organic matrix [147]. The advantages of its use compared to other instrumental examinations are cheapness and easy use of portable instrumentation. It is usually measured on the tibia. The QUS evaluates the speed of sound (SOS) and bone transmission time (BTT). In some studies, to reduce the soft tissue effect, the humerus or metacarpal BTT was preferred over SOS [148,149]. Altuncu et al. observed, through the use of QUS, that there is a decrease in bone mineralization in premature infants in the early postnatal period: the tibial Z-scores of preterm infants (<33 weeks, mean birth weight 1650 g) assessed at term-adjusted age were lower compared with Z-scores at the first week of postnatal life. In this study, it was found that tibial Z-scores in preterm infants of a term-corrected age were inversely proportional to serum ALP levels: in subjects with ALP > 900 IU/L, tibial Z scores were lower than in infants with ALP < 900 IU/L [150].

Rack et al. assessed bone quality using QUS in a cohort of 172 preterm and term infants (gestational age between 23 and 42 weeks; birth weight between 405 and 5130 g): QUS parameters assessed in the first week of life were correlated with gestational age and birth weight [151].

Considering the studies present in the literature, the QUS evaluation for preterm infants may have a significant role in bone health monitoring. However, further studies are needed to identify the biochemical alterations that might correlate better with the QUS parameters [102].

## 7. Prevention and Treatment of Osteopenia

In MBD, prevention plays a more important role than treatment and aims to provide an adequate supply of calcium and phosphorus to promote normal bone development [51].

It is necessary to optimise the diet to prevent MBD, with particular attention to the intake of minerals (calcium, phosphorus) and vitamin D. Individualised mineral supplementation is necessary. It is also advisable, where possible, to limit prolonged exposure to drugs: loop diuretics, methylxanthines, and glucocorticoids [11].

It is important to consider that preterm infants have different bone mineral requirements than full-term newborn infants [118].

Neonatal 25(OH)D concentrations are about 80% of maternal levels, so infants born to vitamin D-deficient mothers are probably also vitamin D deficient. Levels of 25(OH)D in the newborn decrease rapidly as the half-life is about 21 days unless vitamin D supplementation is started immediately with the initiation of enteral feeding. Vitamin D supplementation should be carried out for pregnant women, particularly in countries where vitamin D deficiency is common, in order to ensure sufficient vitamin D levels in the newborn [67,68].

Many groups, including the American Academy of Pediatrics, the Institute of Medicine, the Endocrine Society, and the European Society for Paediatric Gastroenterology, Hepatology, and Nutrition (ESPGHAN), recommend vitamin D supplementation in preterm and term infants [118,152,153,154]

The Institute of Medicine recommendations, based in part on the work of Priemel et al., state that the 25(OH)D level should be 50 nmol/L to achieve good bone mineralization [155].

The Endocrine Society recommends a target level of >75 nmol/L and ESPGHAN > 80 nmol/L [153,154].

Few data are available on the response to vitamin D supplementation in infants weighing < 1200 g at birth [156].

Anderson-Berry et al. carried out a randomized study to evaluate the most appropriate dosage of vitamin D to be administered to preterm infants [156]. The cohort consisted of 32 infants born between 24 and 32 weeks of gestation. The results of this study support the consideration of a daily dose of 800 IU of vitamin D for infants < 32 weeks assisted in the neonatal intensive care unit, resulting in improved 25(OH)D levels at 4 weeks, higher bone density, and improved linear growth [156].

Some neonatologists recommend vitamin D supplementation for premature infants with dosages of 800–1000 IU/day, higher than recommended for term infants (400 IU/day), although there is little evidence to suggest this unless there is hepatic or bowel dysfunction [67,68]. There is no proven benefit to using activated vitamin D (calcitriol or alfacalcidol) versus using the vitamin D parent, except if you have severe kidney or liver disease [67].

During TPN feeding, adequate calcium and phosphate are required and increased during the transition to enteral feedings and during full enteral feedings [120].

Pereira-da-Silva et al. demonstrated that TPN containing high concentrations of calcium and phosphorous during the first weeks of life can prevent bone impairment in preterm infants with an average gestational age of 29.6 weeks and birth weight of 1262 g [120]. Current recommendations in clinical practice are different: for calcium, from 40 to 120 mg/kg/day, and for phosphate, from 30 to 70 mg/kg/day [120]. The introduction of high protein and calorie intakes in TPN in the first days of life, associated with early enteral feeding, have the effect of an increase in cellular uptake of phosphate [157].

To prevent harm from TPN, the transition from total parenteral nutrition to enteral feeding should be accelerated as much as possible [95].

In fact, during the enteral nutrition gastrointestinal, absorption of phosphorus can reach over 90% [158]. Therefore, in the first weeks of life of premature infants, enteral nutrition ensures efficient absorption of calcium and phosphorus [95].

Some studies show that infants fed exclusive breast milk had lower serum phosphorus than those fed with special formulas or taking mineral supplements [159].

Studies show that rickets does occur in 40% of cases of preterm infants fed unenriched breast milk, compared to 16% of those fed special formulas [160].

Fortification remains essential to provide adequate mineral intake: 180 and 200 mL/day of unfortified breast milk probably provide only 1/3 of the level of calcium and phosphorus during growth, despite the fact that the baby absorbs 60% calcium and 80% phosphorus from breast milk [161].

Caution should be exercised when using unfortified donated human milk due to its lower phosphorous content than that of fresh breast milk [11]. Newborns achieve an optimal level of mineral intake with approximately 180–220 mg/kg/day of calcium and 100–130 mg/kg/day of phosphorus [118]. Children with critical illnesses may require targeted mineral supplementation with calcium and/or phosphorus [5].

A recent study in the United Kingdom highlights how phosphate supplementation is considered by neonatologists to be the standard treatment for metabolic bone disease in premature infants. Calcium supplementation is only associated with a smaller number of cases [132].

Phosphorus supplementation should be undertaken if values decrease to <5.5 mg/dL [118].

The recommended dose for phosphorus supplementation is initially 10 mg/kg/day up to 50 mg/kg/day. Individual responses to treatment may vary depending on the clinical state, intestinal pH, absorption, and individual tolerance. The preferred form of phosphorus supplementation is potassium phosphate, intravenous formulation given enterally due to intestinal intolerance of other available phosphate salts. Alternative formulations, such as tablet or powder forms, may also be used. Patients taking potassium-sparing diuretics should be monitored carefully, paying attention to any electrolyte abnormalities, as these alternative formulations contain sodium and potassium [5]. Phosphate supplementation should be undertaken, making sure to maintain a balanced ratio of calcium to phosphate [51].

The ideal ratio of calcium to phosphate intake to optimize phosphate retention and calcium absorption appears to be about 1.5–1.7:1 [118].

In cases of secondary hyperparathyroidism and low TRP, calcium supplementation can be considered [5]. When PTH is elevated, suggesting calcium deficiency, oral calcium supply is recommended to help normalise plasma levels of PTH, phosphate, and ALP. The recommended dose for calcium supplementation is initially 20 mg/kg/day up to 70–100 mg/kg/day [118,133]. This dosage may be increased in preterm infants with severe bone metabolism disorders [51].

There are also non-drug therapies to consider treatment of MBD. Mechanical stimulation is known to play a role in promoting bone growth. Foetal movements performed against the uterine wall help ensure adequate bone mineralization and adequate muscle development [162]. These movements can not be replicated in the perinatal period during the stay in the neonatal intensive care unit. Physical therapy can provide the necessary stimulation with mechanical loading on bones and joints stimulates bone formation and growth [163].

Physiotherapy techniques based on the application of passive movements with light compression have demonstrated favourable effects in preterm infants. On the other hand, the absence of loading increases bone resorption and decreases bone mass [164].

A recent study exploited locomotor reflex therapy (RLT) to produce involuntary active-resistive movements in preterm infants [165].

RLT is an effective technique, more so than other physiotherapy modalities, for promoting bone growth and formation and preventing osteopenia [166]. RLT, through proprioceptive stimuli, produces a central nervous system response causing involuntary, active movements in children [167].

In conclusion, adequate calcium, phosphorus and vitamin D supplementation and a physiotherapy protocol, as well as optimal nutrition, represent the basis for the prevention and treatment of metabolic bone disease in infants.

## 8. Follow-Up Post-Discharge

The newborn baby, therefore, has a much lower skeletal growth rate than the foetus because, regardless of gestational age, transplacental transport is reduced at birth with an immediate decrease in circulating calcium [38].

The level of calcium guaranteed to the foetus during the third trimester is very difficult to achieve “artificially”. However, there are fortificants specifically designed for infants feeding on mother’s milk and some formulated milks with high mineral concentrations that allow bone mineral growth at or close to in utero rates [161].

Fortified breast milk and preterm-specific formulas are generally preferred to formulas with soy or elementary formulas, which lack calcium and phosphorus at levels adequate for a preterm infant [118].

Formulas for premature infants or breast milk fortification are indicated up to a maximum of 52 weeks from the age of conception or, in cases of poor growth, up to 6 months. The duration of fortification is still debated. Infants fed preterm formula, around 6 months of age, achieve bone mineralization comparable to that of full-term infants. Pre-term infants fed with breast milk, on the other hand, reach bone mineralisation comparable to that of full-term infants at around 2 years of age [168].

Decisions on nutritional support are guided by growth parameters, while the frequency of biochemical monitoring is determined by the severity of MBD laboratory parameters.

Particular attention must be paid to adequate supplementation, as excessive doses may cause hypercalciuria and nephrocalcinosis [5]. For this reason, a multidisciplinary approach involving a dietician experienced in neonatal and/or bone health may be useful.

## 9. New Technologies

To date, making an early instrumental diagnosis of MBD is rather complicated. In fact, traditional imaging techniques mainly show frank signs of pathology, such as fractures or significant demineralization (20–40% reduction in mineralization), events that occur at an advanced stage, thus not suitable for early diagnosis [169].

Currently, the first step used by the clinician for the identification of poor bone mineralization is X-ray or QUS, when is possible, even if the gold standard is DEXA.

This technique can be used in both term and preterm infants, in which the lumbar spine, forearm and heel are considered the preferred target regions [170]. However, its use is severely limited by the size of the instrument, the time required to execute images, the presence of moving artifacts, and the impossibility of bedside execution [102].

The use of DEXA is not allowed in pregnancy because it would expose the foetus to ionizing radiation [171,172].

Moreover, DEXA provides quantitative information on bone but does not evaluate qualitative information [144].

A new non-invasive instrumental technique is QUS, which through ultrasound, calculates the bone SOS and BTT evaluating both the bone mineral content and the organic matrix [147,170]

Compared to DEXA, this technique has advantages: low cost, small instrumentation, the possibility of performing the examination at the bedside, and especially the absence of ionizing radiation. However, the results of this technique are operator-dependent and have lower diagnostic accuracy than those provided by DEXA. Therefore, the use of QUS is also severely limited in clinical practice [173,174].

Recently, a new technology known as multi-spectrometry ultrasound spectrometry (REMS) based on ultrasound has been validated. It is very useful in clinical practice for the assessment of bone strength, for the prediction of fracture risk and for the general management of the patient with osteoporosis [144,175].

REMS technology accurately assesses BMD at the femoral neck and lumbar vertebrae, which are considered to be the central anatomical reference sites. During the ultrasound scan, the so-called radio frequency signals are acquired and integrated with ultrasound imaging. Radio frequencies are native unfiltered ultrasound signals, which allow us to obtain the maximum information about the studied tissue, contrary to what happens in the conventional ultrasound-based process where only some signals are processed [144]. Thanks to a fully automatic algorithm, the vast amount of data collected is transformed into a specific spectrum of the patient which is then compared with the reference spectral models by gender, age, site, and BMI extracted from a dedicated database [176,177]. In this way, it is possible to obtain a very accurate estimate of the health of the bone, both quantitatively and qualitatively, that allows us to classify the tissue as healthy, osteopenic, or osteoporotic [144].

The REMS method, as demonstrated by several studies, has a sensitivity and specificity of 90.4% and 95.5%, respectively, compared to DEXA [178], and it is as accurate and reproducible as DEXA in the evaluation of femoral neck BMD [179,180]. The advantages of REMS technology are lower costs, availability in primary service environments without the need for dedicated facilities or certified operators, extreme ease of use thanks to a simple and intuitive acquisition procedure, high rates of reproducibility of the examinations guaranteed by a fast and fully automated data processing and uses ultrasound, safe for foetus [144,175]. All these qualities make REMS the elective technology for extended mass screening [175].

Currently, the use of REMs is not so widespread in clinical practice. In Figure 3, we propose an algorithm that envisages the use of REMs, in the prenatal period, in the identification of newborns at risk and, in the post-natal period, integrating the data obtained with laboratory markers in defining bone mineralisation levels.

## 10. Conclusions

Bone mass is sensitive to many pre- and postnatal insults. Advances in biotechnology and genetics have enabled the identification of numerous microRNAs that contribute to bone development, playing a role in early identifying patients at high risk of developing osteopenia. To implement more comprehensive prevention, it is important to identify not only the factors that have action on bone in the postnatal period but also, where possible, to intervene through strategies in the prenatal period, optimizing maternal nutrition and intrauterine growth. Such strategies should be encouraged as a public health action to reduce the costs associated with the complications of osteopenia. To date, there is a lack of unified guidelines clarifying the correct diagnostic process and management of osteopenic patients. The literature is also lacking in common guidelines for proper prevention. Therefore, there is a need to encourage the use of noninvasive methods in the pre- and postpartum period to implement the best prevention and the creation of new algorithms for the diagnosis and monitoring of bone mineral density.

## Figures and Tables

**Figure 1 nutrients-15-03515-f001:**
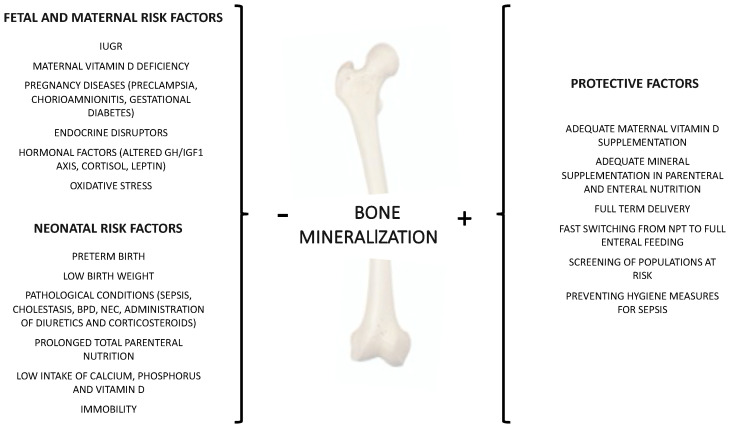
Factors affecting bone mineralization: maternal, foetal, and neonatal risk and protective factors.

**Figure 2 nutrients-15-03515-f002:**
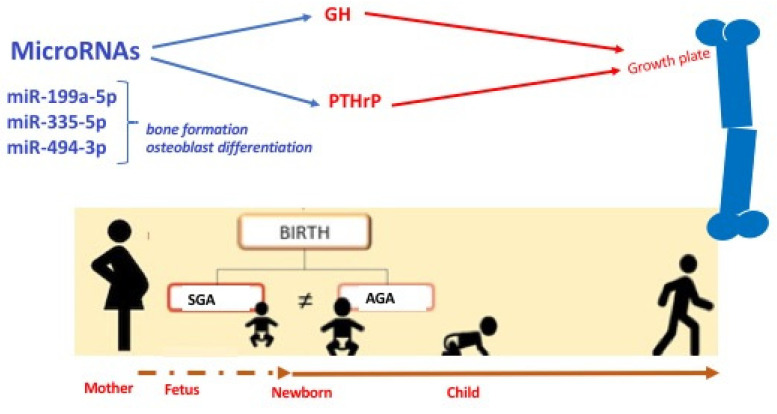
Schematic representation on the role of miRNAs in bone development. miR = microRNA; GH: growth hormone; PTHrP: parathyroid hormone-related protein.

**Figure 3 nutrients-15-03515-f003:**
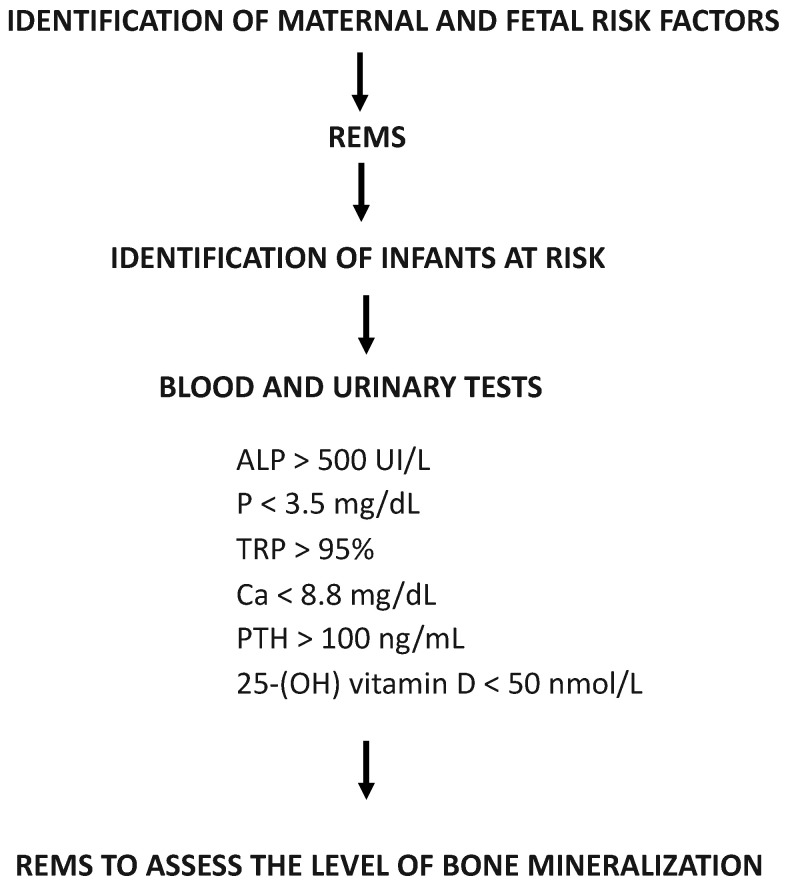
Algorithm proposal: integrated use of laboratory and urinary tests with new technologies for the assessment of bone mineralization levels.

## Data Availability

No data are reported in the paper.

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
