# Peer review of "Prenatal and Neonatal Bone Health: Updated Review on Early Identification of Newborns at High Risk for Osteopenia"

_nutrients, 2023, doi:10.3390/nu15163515_

Round 1
Reviewer 1 Report
Review of the manuscript: Prenatal and neonatal bone health: early identification of 2 newborns at high risk for osteopenia
The work presented deals with a very important topic of population health. The authors have prepared a structured manuscript. They included important data on previously published studies on the topic of bone tissue development. However, I have comments that need clarification. In order to understand the intention and message of the Authors, the manuscript needs to contain specific information: what was the purpose of the manuscript? what hypotheses were made? was it purely a literature review? It is useful to provide study Limitation.
Major Comments:
· The title of the manuscript does not inform whether it is original research, systematic review, literature review?
· The manuscript presented for review has an introduction and several subsections with cited literature from studies by other authors. Upon close reading, the manuscript (if there were a PRISMA protocol) would fit the systematic review or literature review layout. The described contribution of the authors also suggests a review.
· Throughout the paper, there is no clearly defined purpose of this study, no hypotheses, no justification for writing this paper.
· Please specify what was the purpose of preparing this study , what new contributions does it make to science.
· Line 47-54 : “Populations most at risk of developing MBD are: preterm births, infants with low 47 birth weight, particularly very low birth weight (VLBW) and extremely low birth weight 48 (ELBW), infants with intra uterine growth restriction (IUGR), infants with comorbidities 49 typically associated with prematurity like sepsis, cholestasis, bronchopulmonary 50 dysplasia (BPD), necrotizing enterocolitis (NEC), infants requiring long periods of total 51 parenteral nutrition (TPN), infants born to mothers with pregnancy-associated diseases 52 (preeclampsia, chorioamnionitis, gestational diabetes), and infants born to vitamin D-53 deficient mothers.” Comment: no literature reference, needs to be supplemented as the authors state in this paragraph relationships confirmed in studies.
· The conclusion: "Bone is a sensitive organ to numerous pre- and post-natal insults."- This is not a conclusion but a further description of bone tissue knowledge.
Minor points:
· The entire manuscript is worth reading again carefully. Ensure that the analyzed literature items are consistent with the area of your own research.
· The manuscript needs minor technical and stylistic corrections. Large disproportion of the chapters.
· For example Line 121, 129 requires correction of the text formatting
Author Response
Major Comments:
- The title of the manuscript does not inform whether it is original research, systematic review, literature review?
Re: We thank the reviewer for the opportunity to ameliorate our paper.
The title has been changed as follows: Prenatal and neonatal bone health: updated review on early identification of newborns at high risk for osteopenia
- The manuscript presented for review has an introduction and several subsections with cited literature from studies by other authors. Upon close reading, the manuscript (if there were a PRISMA protocol) would fit the systematic review or literature review layout. The described contribution of the authors also suggests a review.
Re: The manuscript aims to be an updated educational review about prenatal and neonatal bone health assessment
- Throughout the paper, there is no clearly defined purpose of this study, no hypotheses, no justification for writing this paper.
Re: Justification to writing the paper and purpose of the review have been now specified at the end of ‘Introduction’ paragraph
- Please specify what was the purpose of preparing this study , what new contributions does it make to science.
Re: The literature supports the increasingly fundamental role of prevention of osteopenia, so it is important to perform accurate management and follow-up of at-risk populations. To date, there is a lack of unified guidelines clarifying the correct diagnostic process and management of osteopenic patients. This review contribute to update the knowledge related to bone health and to encourage the use of non invasive tools (ultrasounds) to monitor bone mineral density from the earliest stage of life, laying the foundations for the study of new biomarkers of osteopenia that take into account the environmental interference on bone health.
These concepts have been clearly stated in the revised manuscript
- Line 47-54 :“Populations most at risk of developing MBD are: preterm births, infants with low 47 birth weight, particularly very low birth weight (VLBW) and extremely low birth weight 48 (ELBW), infants with intra uterine growth restriction (IUGR), infants with comorbidities 49 typically associated with prematurity like sepsis, cholestasis, bronchopulmonary 50 dysplasia (BPD), necrotizing enterocolitis (NEC), infants requiring long periods of total 51 parenteral nutrition (TPN), infants born to mothers with pregnancy-associated diseases 52 (preeclampsia, chorioamnionitis, gestational diabetes), and infants born to vitamin D-53 deficient mothers.” Comment: no literature reference, needs to be supplemented as the authors state in this paragraph relationships confirmed in studies.
Re: Reference n.5 has been added at the end of the paragraph
- The conclusion: "Bone is a sensitive organ to numerous pre- and post-natal insults."-This is not a conclusion but a further description of bone tissue knowledge.
Re: We thank the reviewer for this remark. The phrase has been removed.
Minor points:
- The entire manuscript is worth reading again carefully. Ensure that the analyzed literature items are consistent with the area of your own research.
Re: the text has been carefully checked, in order to verify consistency of literature with the area of research
- The manuscript needs minor technical and stylistic corrections. Large disproportion of the chapters. For example Line 121, 129 requires correction of the text formatting
Re: the text has been carefully checked and corrected

Reviewer 2 Report
The review paper titled "Prenatal and Neonatal Bone Health: Early Identification of Newborns at High Risk for Osteopenia," authored by Perrone et al., provides an extensive analysis of the current evidence regarding the early identification and management of populations at high risk for osteopenia. The authors also discuss the implementation of screening and prevention programs based on factors such as gestational age, weight, and morbidity.
Comments:
1. It is worth noting that the explanation of the mechanism through which the intrauterine environment affects bone health lacks clarity.
2. It would be beneficial for the authors to elucidate this mechanism by visually depicting the signaling molecules or pathways involved in prenatal and neonatal bone development.
3. Additionally, the authors should explore potential therapeutic approaches or interventions that target these pathways to mitigate osteopenia.
Author Response
The review paper titled "Prenatal and Neonatal Bone Health: Early Identification of Newborns at High Risk for Osteopenia," authored by Perrone et al., provides an extensive analysis of the current evidence regarding the early identification and management of populations at high risk for osteopenia. The authors also discuss the implementation of screening and prevention programs based on factors such as gestational age, weight, and morbidity.
Comments:
- It is worth noting that the explanation of the mechanism through which the intrauterine environment affects bone health lacks clarity.
Re: We thank the reviewer for the opportunity to ameliorate our paper. a paragraph on the influence of endocrine disruptors on bone development has been better clarified and added in the text - It would be beneficial for the authors to elucidate this mechanism by visually depicting the signaling molecules or pathways involved in prenatal and neonatal bone development.
Re: A new figure (Figure 2) has now provided.
- Additionally, the authors should explore potential therapeutic approaches or interventions that target these pathways to mitigate osteopenia.
Re: We thank the reviewer for this remark. Potential therapeutic approaches or interventions that target these pathways has been provided in the new paragraph added to the manuscript (lines 347-391)

Round 2
Reviewer 2 Report
The authors have addressed all the major questions raised and provided satisfactory responses. I recommend that the manuscript be deemed acceptable for publication in the journal Nutrients.